# Endoscopic Delivery of Polymers Reduces Delayed Bleeding after Gastric Endoscopic Submucosal Dissection: A Systematic Review and Meta-Analysis

**DOI:** 10.3390/polym14122387

**Published:** 2022-06-13

**Authors:** Youli Chen, Xinyan Zhao, Dongke Wang, Xinghuang Liu, Jie Chen, Jun Song, Tao Bai, Xiaohua Hou

**Affiliations:** 1Division of Gastroenterology, Union Hospital, Tongji Medical College, Huazhong University of Science and Technology, Wuhan 430022, China; chenylpetri@gmail.com (Y.C.); wdkeee@126.com (D.W.); xh_liu@hust.edu.cn (X.L.); jie_chen_33@163.com (J.C.); houxh@hust.edu.cn (X.H.); 2Department of Spleen and Stomach Disease, Hubei Provincial Hospital of Traditional Chinese Medicine, Wuhan 430006, China; 15827576532@163.com

**Keywords:** endoscopic submucosal dissection, delayed bleeding, endoscopic closure, polyglycolic acid, hemostatic spray, polymers

## Abstract

New endoscopic approaches for the prevention of delayed bleeding (DB) after gastric endoscopic submucosal dissection (ESD) have been reported in recent years, and endoscopic delivery of biodegradable polymers for iatrogenic ulcer hemostasis and coverage has emerged as one of the most promising techniques for post-ESD management. However, the comparative efficacy of these techniques remains uncertain. We performed a systematic search of multiple databases up to May 2022 to identify studies reporting DB rates as outcomes in patients undergoing gastric ESD who were treated with subsequent endoscopic management, including endoscopic closure (clip-based methods and suturing), PGA sheet tissue shielding, and hemostatic powder/gel spray (including polymeric sealants and other adhesives). The risk ratios (RRs) of delayed bleeding in treatment groups and control groups were pooled, and the Bayesian framework was used to perform a network meta-analysis (NMA). Among these studies, 16 head-to-head comparisons that covered 2742 lesions were included in the NMA. Tissue shielding using PGA sheets significantly reduced the risk of DB by nearly two thirds in high-risk patients, while hemostatic spray systems, primarily polymer-based, reduced DB in low-risk patients nine-fold. Researchers should recognize the essential role of polymers in the management of ESD-induced ulcers, and develop and validate clinical application strategies for promising materials.

## 1. Introduction

Endoscopic submucosal dissection (ESD) has become the standard procedure for the treatment of many gastrointestinal lesions, including gastric superficial neoplastic lesions, due to its minimal invasiveness and high rate of en bloc resection [1]. However, ESD is a complex procedure that requires considerable endoscopic skills and has a relatively high potential for serious adverse events, such as delayed bleeding, which may lead to potentially severe consequences, such as hemorrhagic shock [2]. There are currently two established effective bleeding prevention methods: proton pump inhibitors (PPI) [3,4], and coagulation or clipping of visible vessels in post-ESD ulcers [5]. However, even with these preventive methods, the rate of post-ESD bleeding is approximately 5% [6,7,8,9], signifying that post-ESD bleeding cannot be completely prevented with only these standard methods of care. Post-ESD bleeding is becoming ever more prominent since the population of patients taking antithrombotic agents is increasing and the indications for ESD are expanded [10]. Although other factors are still controversial, the usage of antithrombotic agents and large resection size are known to be significant risk factors for post-ESD bleeding [11], with post-ESD bleeding rates as high as 21–38% in populations with these characteristics [12,13,14].

For the above reasons, the development and validation of effective prevention methods for post-ESD bleeding in gastric lesions are desirable. Closure of resection-induced ulcers of the stomach with endoscopic clips has been performed for many years, and endoscopic suturing techniques for defect closure have also been developed and applied in clinical settings. In recent years, the endoscopic application of biodegradable polymers has emerged as one of the most promising anti-bleeding techniques for post-ESD management. Polyglycolic acid (PGA), a synthetic, braided polymer commonly used in surgical procedures, has been combined with fibrin, a natural biopolymer, to create a much stronger sealant than any other biomaterial combination for GI ulcer coverage [15,16,17]. In addition, endoscopic delivery of hemostats has shown potential benefits for the prevention of delayed bleeding, despite its short retention period [18,19]. Among these hemostatic sprays, polymer-based hemostatic systems, such as EndoClot (polysaccharide particle powder), Surgicel (polyanhydroglucuronic acid gauze), and PuraStat (peptide solution) have received FDA approval or premarket approval as hemostats. These preventive effects of endoscopic treatments for post-ESD delayed bleeding could be due to several mechanisms. Firstly, they protect ulcer surfaces from the physical stimulation of food and the chemical stimulation of digestive enzymes and gastric acid. Secondly, specific hemostatic materials aid hemostasis by lowering the pH of tissue surrounding the ulcer and causing vasoconstriction, accelerating the physiologic clotting system, or providing a substrate for platelet adsorption and aggregation.

However, clinical outcomes and preliminary research regarding endoscopic approaches for the prevention of post-ESD delayed bleeding in gastric lesions were inconclusive and controversial. To date, evidence and data on the efficacy of these procedures have not been systematically reviewed and compared. Thus, this study aimed to review the efficacy of preventive methods used for the prevention of delayed bleeding after gastric ESD, which could be beneficial for establishing sound treatment options, and direct future research to develop and expand the usage of biomedical materials suitable for endoscopic application.

## 2. Materials and Methods

### 2.1. Data Sources and Search Strategy

This protocol was registered and published in PROSPERO on 21 May 2022 [ID: CRD42022331772]. We performed a comprehensive literature search by using Medline, EMBASE, and the Cochrane Library (up to May 2022) to identify full articles evaluating outcomes of endoscopic management (e.g., tissue shielding (PGA sheets with/without combination of fibrin glue), endoscopic closure (endoclips, endoloops, over-the-scope-clip [OTSC], and suturing, etc.), and hemostatic spray (polymeric sealants and other adhesives)) for the prevention of delayed bleeding after gastric ESD. Electronic searches were supplemented by manual searches of references of included studies and review articles. Specific search strategies are available in Appendix B.

### 2.2. Selection Process

Two reviewers (Y.C., X.Z.) independently screened the titles and abstracts of articles and reviewed the full text of any title or abstract deemed potentially eligible by either reviewer. Disagreements were resolved through discussion between reviewers. The reasons for excluding trials were recorded. Neither of the review authors was blinded to the journal titles, or the study authors or institutions. When there were multiple articles for a single study, reviewers used the latest publication.

### 2.3. Data Extraction

By using standardized forms, 2 reviewers (Y.C., X.Z.) extracted data independently and in duplicate from each eligible study. Reviewers resolved disagreements by discussion. The reviewers extracted the following data from each study: study characteristics (study design and location, number of centers involved, follow-up time), patient characteristics (antithrombotic use, large resection size), endoscopic procedures, and delayed bleeding rates. The procedure time was also extracted if provided.

### 2.4. Quality Assessment

The Newcastle–Ottawa scale [20] was used to assess the quality of individual cohort and single-arm studies. In brief, a maximum of 9 points was assigned to each study: 4 for selection, 2 for comparability, and 3 for outcomes. Scores of ≥5, 3 to 4, and ≤2 were considered to be indicative of a high-quality, medium-quality, and low-quality study, respectively [21]. The Cochrane Risk of Bias assessment tool [22] and the Risk Of Bias In Non-randomized Studies of Interventions (ROBINS-I) assessment tool [23] were used for randomized controlled trials (RCT) and non-randomized studies, separately. Case studies were appraised according to Murad et al. [24] Two reviewers (Y.C., X.Z.) assessed quality measures for the included studies, and discrepancies were adjudicated by collegial discussion.

### 2.5. Inclusion and Exclusion Criteria

For this systematic review and network meta-analysis, studies were considered eligible if they met the following criteria: (1) RCTs, prospective and retrospective non-RCTs, and case studies on human subjects regarding endoscopic management for prevention of delayed bleeding after gastric ESD; (2) the risk ratio (RR) and the corresponding 95% confidence interval (CI) of delayed bleeding were reported or could be calculated through the sufficient data provided; (3) published as full text. Studies were excluded if they were (1) not published in the English language, (2) conference abstracts, and (3) case reports.

### 2.6. Outcome Assessment

The primary outcome of the systematic review and network meta-analysis was the post-ESD delayed bleeding rate. The secondary outcome was the procedure time required for each post-ESD anti-bleeding technique.

### 2.7. Statistical Analysis

First, to harmonize data from the noncomparative cohorts pooled, a meta-analysis of proportions of delayed bleeding was conducted. Next, a direct pair-wise meta-analysis of trials that compared different treatments was conducted using random-effects models. For each pairwise comparison of the dichotomous outcome, pooled data were expressed as risk ratio (RR) and 95% CI. Then, the network meta-analysis (NMA) model was estimated with random-effects models based on a Bayesian framework and Markov Chain Monte Carlo (MCMC) theory to incorporate the estimates of direct and indirect intervention comparisons [25]. Moreover, all the treatments were ranked based on the analysis of ranking probabilities.

Heterogeneity was visualized with forest plots and quantified using the I^2^ statistic. Adjectives of low, moderate, and high were assigned to I^2^ values of 25%, 50%, and 75%, respectively. We also tested heterogeneity using the Q test (statistical significance level set as *p* < 0.1).

Additionally, subgroup analysis was performed regarding bleeding risk stratification. High-risk patients were defined as those with a large resection owing to a specimen size ≥ 40 mm, or those continuing antithrombotic agents. The value of deviance information criterion (DIC) of random-effects and fixed-effects models was used for the selection of models. The smaller the DIC is, the better the model; a DIC difference within 5 indicates the models’ competitiveness, and a difference greater than 10 can rule out the model with the higher DIC value [26]. Random-effects models were used unless fixed-effects models showed significant superiority in model fit over the former (DIC > 10). The publication bias was assessed by evaluating a funnel plot of the standard error of the treatment estimates for asymmetry. The symmetry of funnel plots was assessed visually, with Egger’s test [27], and with the adjusted rank correlation test [28].

Furthermore, we pooled the mean procedure time of each post-ESD anti-bleeding management technique using a meta-analytical approach, similar to that used in the non-comparative trial synthesis of delayed bleeding rates. To harmonize data, medians and ranges or interquartile ranges were transformed into means and standard deviations, using the estimation method according to Wan et al. [29]. Procedure time required for anti-bleeding techniques was calculated as mean difference if complete ESD procedure time was reported in the original article. All calculations were performed using packages from the open-source software environment R Studio, version 1.4.12.

## 3. Results

### 3.1. Identified Studies and Quality

The Preferred Reporting Items for Systematic Reviews and Meta-Analyses (PRISMA) reporting guideline was followed to conduct the study [30] (Appendix C). Study selection procedures are illustrated in Figure 1. Searches of three primary electronic databases identified 618 unique abstracts, titles, or both identified as original publications. Of the total, 54 proved potentially relevant for full-text review. Of these, 31 articles on the management of ESD-induced ulcers via endoscopic closure (hemoclips, endoclips, endoloop + endoclips, detachable snare + clips, OTSC + through-the-scope clips [TTSC], overstitch suturing, handsewn suturing) [31,32,33,34,35,36,37,38,39,40,41,42], tissue shielding methods (PGA sheets-based) [10,43,44,45,46,47,48,49], and hemostatic powder/gel spray [18,19,50,51,52,53,54,55,56,57,58], matched the selection criteria and were included in the systematic review. The key characteristics of each included study appear in Table 1. Among these studies, 23 reported data relevant to procedure time for anti-bleeding techniques. Seventeen studies were performed in Japan, and the remainder were performed in Korea, China, France, the United States, the United Kingdom, and Malaysia. There were six RCTs, and all but five studies were single-centered. These studies were all included in the non-comparative trial synthesis. Of the 31 studies, 15 were excluded because they only included single-arm trials; finally, 16 studies were included in the comparative trial synthesis, including pair-wise and network meta-analysis.

The details of the quality of study assessment are summarized in Appendix A. We used the Newcastle–Ottawa Scale for cohort studies to assess the methodological quality of cohort and single-arm studies; the scores ranged from five to nine, meaning that all were of of high quality. Tsuji et al. [49] was at moderate risk of bias, evaluated using the ROBINS-I assessment tool. Overall, RCTs assessed with the Cochrane Risk of Bias assessment tool were considered to be at unclear risk of bias, with most information from studies at low or unclear risk of bias. Case series were of adequate quality for non-comparative trial synthesis.

### 3.2. Non-Comparative Trial Synthesis

#### 3.2.1. Delayed Bleeding Rates

We examined the three endoscopic approaches’ effects for the prevention of delayed bleeding in 31 studies, included the non-comparative trial synthesis (Appendix A). The total numbers of cases included in the endoscopic closure, tissue shielding, hemostatic spray, and control group were 414, 322, 678, and 1678, respectively. The total delayed bleeding rates after endoscopic closure, tissue shielding, and hemostatic spray were 0.02 (95% CI 0.00–0.06), 0.05 (95% CI 0.03–0.08), and 0.01 (95% CI 0.00–0.04), respectively, while that of those treated with conventional approaches was 0.09 (95% CI 0.06–0.13). Additionally, among endoscopic closure methods, suturing slightly outperformed clip-based approaches (Appendix A).

Subgroup analysis stratifying bleeding propensity indicated that high-risk patients undergoing any of anti-bleeding management, namely endoscopic closure (0.05 [95% CI 0.00–0.14]), tissue shielding (0.04 [95% CI 0.02–0.08]), and hemostatic spray (0.07 [95% CI 0.03–0.12]) experienced much less delayed bleeding incidence compared with those treated with conventional methods (0.14 [95% CI 0.09–0.20]) (Appendix A); while low-risk patients (0.06 [95% CI 0.03–0.10]) benefited from endoscopic closure (0.03 [95% CI 0.00–0.09]) and hemostatic spray (0.01 [95% CI 0.00–0.04]) (Appendix A).

#### 3.2.2. Procedure Time

The pooled procedure times required for endoscopic closure methods, including endoscopic hand suturing (EHS), endoscopic overstitch suturing, and clip-based methods were 19.36 (95% CI 14.86–24.06) minutes, 42.02 (95% CI 34.35–49.69) minutes, and 12.65 (95% CI 9.89–15.41) minutes, respectively. The procedure time for tissue shielding was only second to EHS, averaging 27.39 (95% CI 17.34–37.45) minutes. Performing hemostatic spray procedures took considerably less time than the above methods, with a pooled mean of 5.54 min (Appendix A).

### 3.3. Pairwise Meta-Analysis Results

The pooled RR across studies comparing tissue shielding with control groups was 0.45 (95% CI 0.25–0.82), with a negligible level of heterogeneity (I^2^ = 0%, *p* = 0.45), which indicates tissue shielding significantly reduces the risk of delayed bleeding in post-ESD patients. Post-ESD management using hemostatic spray also showed significant efficacy in reducing delayed bleeding risk (RR 0.46, 95% CI 0.25–0.83; I^2^ = 16%, *p* = 0.31). However, in studies comparing the efficacy of prevention of delayed bleeding using endoscopic closure (RR 0.60, 95% CI 0.27–1.32; I^2^ = 43%, *p* = 0.150) with control groups, no statistically significant difference was found (Figure 2).

Next, subgroup analysis was performed. Overall, we identified 1196 lesions in eight studies from high-risk patients, and 1423 lesions in eight studies from low-risk patients. Results showed that high-risk patients benefited from receiving tissue shielding treatment using PGA sheets compared with conventional methods (RR 0.35, 95% CI 0.17–0.69; I^2^ = 0%, *p* = 0.63) (Appendix A), while no methods showed significant efficacy in preventing delayed bleeding in low-risk patients (Appendix A).

### 3.4. Network Meta-Analysis Results

#### 3.4.1. Network Quality

All 16 studies included in the network meta-analysis were two-armed; among these, four, six, and six studies, respectively, compared the delayed bleeding rates between the group that received post-ESD endoscopic closure, tissue shielding, and hemostatic spray, and the group treated with conventional coagulation and/or clipping of the visible vessels. Thus, a network was created among the nodes of the three categorized endoscopic approaches (endoscopic closure, tissue shielding method, and hemostatic spray) with a control group node (Figure 3A). Overall, 2742 lesions were randomly assigned to one of the three endoscopic approaches for the prevention of delayed bleeding or the conventional treatment, and were included in the network meta-analysis.

The heterogeneity of the network meta-analysis was moderate across comparisons (Global I^2^ = 36.99%). There were no comparisons to assess for inconsistency [59], and the random-effects model was selected based on DIC evaluation (fixed-effects model, DIC: 59.74; random-effects model, DIC: 58.835).

#### 3.4.2. Delayed Bleeding Rates

Appendix A and Figure 4A show the delayed bleeding comparison results. Tissue shielding (RR 0.37, 95%; CI 0.15–0.88; Global I^2^ = 0%) and hemostatic spray (RR 0.34, 95%; CI 0.13–0.77; Global I^2^ = 43.65%) significantly reduces the risk of delayed bleeding in control group post-ESD patients. Non-significant results were found when comparing the rest of the approaches. As for ranking the overall results, the efficacy of endoscopic approaches for the prevention of delayed bleeding, in order of most to least, are as follows: hemostatic spray, tissue shielding methods, endoscopic closure, and conventional coagulation and clipping of visible vessels (Appendix A).

#### 3.4.3. Subgroup Network Analysis

Subgroup analysis according to bleeding risk was conducted to investigate whether post-ESD endoscopic treatment could benefit high-risk or low-risk patients (Figure 3B,C).

The heterogeneity the network meta-analysis in high-risk patients (N = 1196) was null across comparisons (Global I^2^ = 0.06%). Similarly, there were no comparisons to assess for inconsistency, and we selected the random-effects model (fixed-effects model, DIC: 26.87; random-effects model, DIC: 27.64). Only tissue shielding significantly reduces the risk of delayed bleeding in this population (RR 0.32, 95% CI 0.12–0.79; Global I^2^ = 0%). Non-significant results were found when comparing the rest of the treatments (Appendix A, Figure 4B). The efficacy of endoscopic approaches, in order of most to least, are as follows: tissue shielding methods, hemostatic spray, endoscopic closure, and conventional coagulation and clipping of visible vessels (Appendix A).

In lesions in low-risk patients (N = 1423), this subgroup network meta-analysis showed a high level of heterogeneity (Global I^2^ = 80.89%). DIC evaluation (fixed-effects model, DIC: 39.26; random-effects model, DIC: 28.93) indicated that the random-effects model greatly improved model fit. The efficacy of hemostatic spray for iatrogenic ulcers was established in low-risk patients (RR 0.11, 95% CI 0.01–0.87; Global I^2^ = 81.52%) (Appendix A, Figure 4C). Hemostatic spray, followed by tissue shielding methods, and endoscopic closure, is potentially beneficial for the prevention of delayed bleeding in this population, according to ranking results (Appendix A).

#### 3.4.4. Publication Bias

All statistical tests of funnel plot asymmetry were non-significant, implying a low risk of publication bias (Appendix A).

## 4. Discussion

Developing approaches to prevent delayed bleeding after gastric ESD significantly reduces the duration of hospital stay, and could greatly relieve the economic burden of patients. To our knowledge, this is the first systematic review and network meta-analysis regarding the efficacy of endoscopic management for the prevention of delayed bleeding after gastric ESD. Given that the number of patients enrolled in individual trials rarely exceeds 100, this meta-analysis non-comparatively synthesized data from 31 studies, and pooled data in 16 studies for pairwise and network meta-analysis, yielding a much larger sample size and resulting in adequate statistical power. This study could provide the endoscopic research community with sensible estimates on the efficacy of endoscopic approaches for the prevention of delayed bleeding in gastric ESD. This NMA also complements current literature by indirectly comparing endoscopic approaches, and generating efficacy rankings of these techniques, since no head-to-head trial was conducted to compare the efficacy of different techniques.

According to our NMA, in patients with mixed risk levels for delayed bleeding, tissue shielding methods using PGA sheets and hemostatic spray both significantly reduce the risk of delayed bleeding by approximately two thirds, compared with untreated ESD-induced ulcers. PGA application was mostly studied in high-risk patients, and a similar anti-bleeding effect size was found in this population. Other methods, however, were found to be of insignificant efficacy in this group of patients. In low-risk patients, hemostatic spray ranked first among the potentially beneficial approaches, and reduced the delayed bleeding risk nine-fold, albeit with considerable heterogeneity among studies. Overall, our study suggests tissue shielding with PGA is most promising for the prevention of post-ESD delayed bleeding in high-risk patients, and hemostatic spray is likely to benefit those with a low risk of bleeding. Taking into account the reasonable procedure time required for their application, polymeric materials for hemostasis and coverage of ulcer surfaces greatly expands the therapeutic toolbox available for the effective prevention of post-ESD adverse events.

A recent meta-analysis carried out by Li and colleagues [60] concluded that PGA sheets effectively reduced the post-ESD bleeding rate in patients receiving antithrombotic agents (RR 0.39; 95% CI, 0.18–0.83). In our study, with systematic search updated to May 2022, and including data from Kikuchi et al. [46] and Abiko et al. [43], the results were similar to that of Li’s work, further validating the potential of endoscopic application for post-ESD bleeding prevention. PGA was one of the initial, degradable polymers researched for biomedical application, and its use has now been extended to wound healing and adhesives for soft tissues [17]. PGA combined with fibrin sealant has been proven to be a powerful adhesive, now commonly used in oral surgery [61]. PGA provides abundant cytoskeletons to support cell crawling during the repair process, inspires epithelization, inhibits rejection reactions due to its strong degradative function, and reduces inflammatory responses [62,63]. It is highly biocompatible in most of its applications, although acute inflammation induced by degraded PGA has also been reported [64]. In the scenario of post-ESD ulcer coverage, Murakami also suggests that early bleeding PGA-shielded post-ESD ulcers could produce hemostasis difficulties, because the degenerated PGA sheet obstructs accurate identification of the bleeding point [65]. Moreover, producing PGA sheets combined with fibrin glue is costly, amounting to approximately $500 [66], and the application procedure is relatively time-consuming and demanding. Nevertheless, the application of PGA sheets for the prevention of delayed bleeding after gastric ESD seems particularly promising, as related studies advancing this technique are rapidly accumulating. A novel device delivery station system (DDSS) is also being developed for rapid and tight PGA delivery and affixation [67]. In a recent single-arm study, Kobayashi and colleagues [47] proposed a wafer paper and ring-mounted PGA sheet (WaRP) method, which required 10.5 ± 6.7 min for fixation, the fastest to date.

Hemostatic powder/gel spray is likely to prevent post-ESD delayed bleeding in low-risk patients. The trials included in our study predominantly investigated the efficacy of hemostatic spray using natural or synthetic biodegradable, organic polymer-based materials, except for a mineral hemostatic powder TC-325 [50], and an α-cyanoacrylate medical adhesive [57]. Although hemostatic powder/gel typically act on bleeding within two to three days, it could potentially prevent delayed bleeding by accelerating ulcer healing. Polyethylene oxide granule spray, as used in Yu’s study [56], and polysaccharide powder, as used in a couple of studies [51,53,68], could form a gelled layer, which not only mechanically protects the wounds but also accelerates the physiological coagulation process, thus speeding up the formation of blood clots. Purastat, a transparent self-assembling peptide gel, has been proposed to effectively form a protective mucosal barrier, and facilitate ulcer healing judged by the transitional rates of the healing and scarring stages [58]. Becker and his team also found that fibrin glue positively modulates ulcer healing by causing an increased number of proliferating cells in the ulcer margin, and enhancing the density of microvessels [69]. One of the advantages of hemostatic spray is that its use requires minimal technical expertise and procedure time. In this light, hemostatic spray for ESD-induced ulcers is relatively simple and reliable, and could potentially be a widespread method. Hemostatic polymers, especially, could play an essential role in the management of post-ESD complications.

Although endoscopic closure of ESD-induced ulcers could potentially facilitate mucosal wound healing [34] and has been practiced for many years, our study finds its preventative efficacy less than optimal in patients undergoing ESD. Clip-based endoscopic closure is limited to certain areas and lesion types. For example, ulcers in the cardiac or pyloric region could not be closed using a combination of a single OTSC and TTSC for a high risk of stenosis after ESD, and complete closure of extensive lesions is difficult [39]. Although EHS [35] and overstitch suturing [37] may be feasible approaches even for a large mucosal defect, these reports were only a few case series with small sample sizes and were only included in our non-comparative trial synthesis. Additionally, the average procedure time of EHS is over 40 min, the longest of the methods we investigated, whereas the overstitch device amounts to over $2500 [70]. The cost-effectiveness of endoscopic closure approaches needs to be determined by conducting more trials and collecting more data in the future.

Our study has several limitations. Firstly, no study in our network meta-analysis directly compared different endoscopic approaches, so the inconsistency of the network cannot be assessed. Secondly, the present literature is still subject to a relatively small sample size and a small number of studies, and we were unable to assess possible confounding factors for preventive efficacy. Thirdly, although the efficacy of endoscopic treatments of ESD-induced ulcers was confirmed in our study, questions remain regarding the utility and feasibility of these techniques, limiting the impact of such data.

After conducting a systematic review of the existing evidence, we conclude that endoscopic PGA shielding methods are beneficial for the prevention of delayed bleeding after gastric ESD, especially in high-risk patients, while hemostatic spray would suffice for anti-bleeding management in low-risk patients. Future studies should also focus on developing technically feasible and cost-effective materials and approaches for the prevention of ESD-related adverse events. Multi-center RCTs that directly compare different endoscopic approaches with strict inclusion criteria and adequate follow-up periods should be conducted in the future to acquire more clinical evidence.

## Figures and Tables

**Figure 1 polymers-14-02387-f001:**
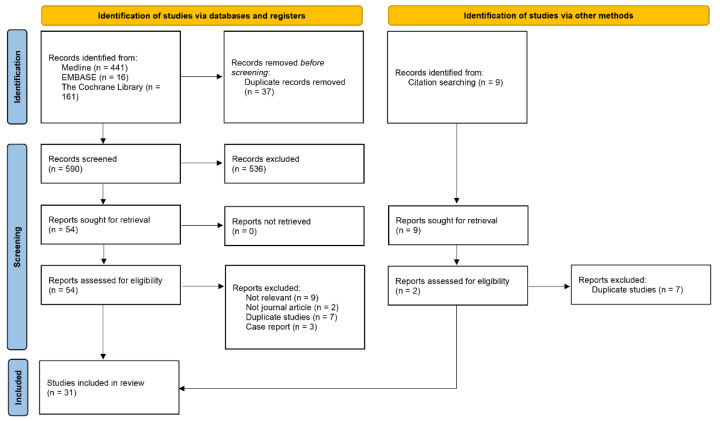
Study identification and selection.

**Figure 2 polymers-14-02387-f002:**
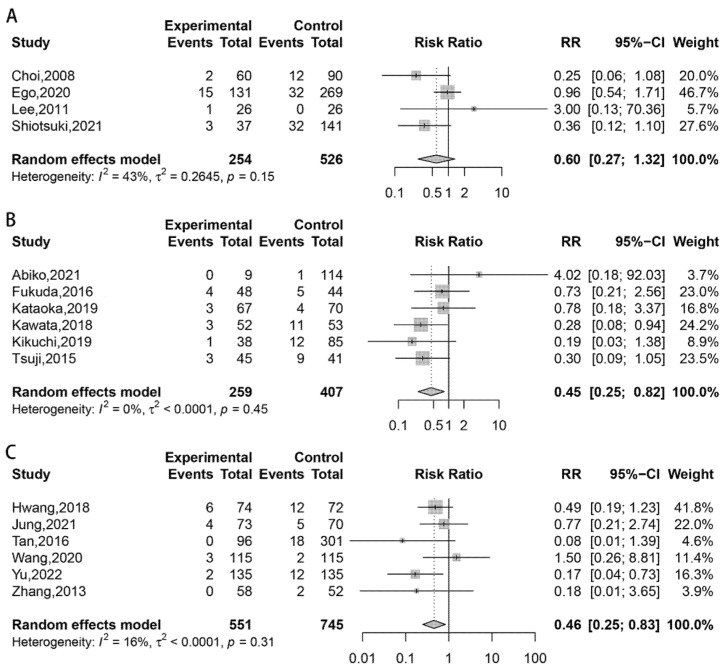
The efficacy of different endoscopic approaches for the prevention of delayed bleeding after gastric ESD, according to pair-wise meta-analysis: (**A**) Endoscopic closure group vs. control group [32,33,38,41]; (**B**) Tissue shielding group vs. control group [10,43,44,45,46]; (**C**) Hemostatic spray group vs. control group. Block and whisker: point estimate and 95% confidence interval (CI) of the primary study. Its relative size and proximity to the meta-analysis pooled estimate are proportional to primary study relative weight. Grey diamond: Pooled estimate of effect size. Its width corresponds to its 95% CI [19,52,53,55,56,57].

**Figure 3 polymers-14-02387-f003:**
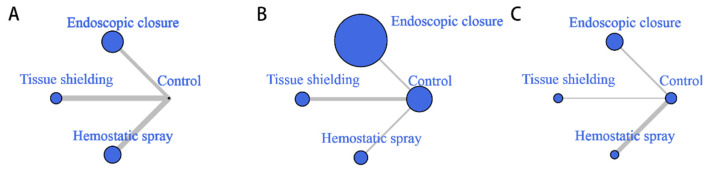
Network graph of the included studies: (**A**) in patients overall; (**B**) in high-risk patients; (**C**) in low-risk patients. The size of the node is proportional to the number of participants in the group (in (**A**), as the number of patients in the control group is much larger than the number in other groups, the control group node size is collapsed for graph neatness), and the width of the edge is proportional to the number of studies comparing two approaches.

**Figure 4 polymers-14-02387-f004:**
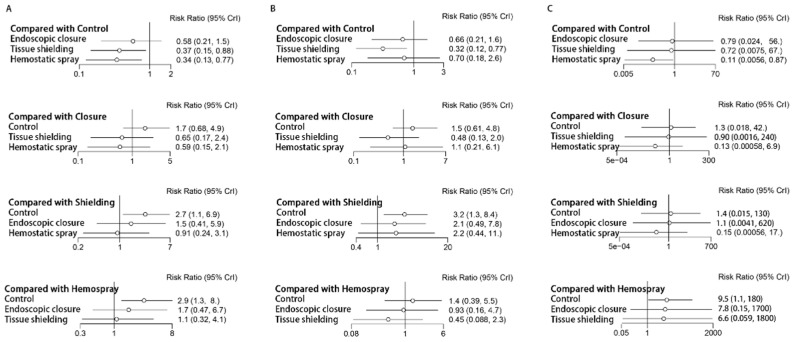
The efficacy of different endoscopic approaches for the prevention of delayed bleeding after gastric ESD, according to network meta-analysis: (**A**) in patients overall; (**B**) in high-risk patients; (**C**) in low-risk patients.

**Table 1 polymers-14-02387-t001:** Study characteristics.

Article (Author, Year)	Country	Design	No. of Centers	Sample Size	Patient Characteristics	Additional Procedure Time (Mean ± SD)	Study Group Intervention	Control Group Intervention	Follow-Up Time
Abiko, 2021 [43]	Japan	Retrospective case series	single	123	High risk in treatment group and low risk in control group	55.33 ± 16.62	Modified search, coagulation, and clipping method (HX-610-135S; Olympus, Tokyo, Japan) + PGA (Neoveil; Gunze Co., Tokyo, Japan) + fibrin glue (Beriplast P combi-set; CSL Behring Pharma, Tokyo, Japan) (PMSCC)	Modified search, coagulation, and clipping (HX-610-135S; Olympus, Tokyo, Japan) method (MSCC)	7 days
Akimoto, 2021 [31]	Japan	Prospective, single-arm study	single	20	High risk	38.36 ± 10.56	Endoscopic hand suturing (EHS) (VLOCL0604; Covidien, Mansfield, MA, USA)	NA	4 weeks
Choi, 2008 [32]	Korea	Retrospective cohort study	single	150	Low risk	18.00 ± 7.78	Hemoclips (HX600-090L; Olympus, Tokyo, Japan)	Heat probe coagulation, coagulation forceps, argon plasma coagulation (APC), and/or hemoclips (HX600-090L; Olympus, Tokyo, Japan)	2 days
Ego, 2020 [33]	Japan	Retrospective cohort study	single	400	High risk	23.50 ± 10.80	Endoloop (MAJ-254; Olympus Medical, Tokyo, Japan) and Endoclips (HX-610-090, Olympus Medical, Tokyo, Japan or ZEOCLIP ZP-CH, Zeon medical, Tokyo, Japan)	Coagulation using hemostatic forceps	56 days
Fukuda, 2016 [44]	Japan	Retrospective cohort study	single	92	Low risk	NA	PGA (Neoveil; Gunze Co., Tokyo, Japan) + fibrin glue (Beriplast P combi-set; CSL Behring Pharma, Tokyo, Japan); modified clip-and-pull method	Non-sealing	≥40 days
Goto, 2020 [35]	Japan	Prospective, single-arm study	multiple	30	Mixed and grouped	46.20 ± 17.00	EHS (VLOCL0604; Covidien, Mansfield, MA, USA)	NA	3–4 weeks
Goto, 2017 [34]	Japan	Prospective case series	single	18	NA	NA	EHS (VLOCL0604; Covidien, Mansfield, MA, USA)	NA	4 weeks
Haddara, 2016 [50]	France	Retrospective case series	multiple	2	Mixed	NA	TC-325 hemostatic powder (Hemospray; Cook Medical, Winston-Salem, NC, USA)	NA	30 days
Hahn, 2017 [51]	Korea	Prospective, single-arm study	single	44	High risk	NA	Polysaccharide hemostatic powder (EndoClot; Endo-Clot Plus, Inc., Santa Clara, CA, USA)	NA	4 weeks
Han, 2020 [36]	USA	Prospective cohort study	single	18	Mixed	13.40 ± 5.90	Endoscopic overstitch suturing (Apollo Endosurgery Inc., Austin, TX, USA)	NA	6 months
Hwang, 2018 [52]	Korea	RCT	single	146	Low risk	NA	Polyanhydroglucuronic acid gauze (Surgicel; Ethicon Inc., Johnson and Johnson, Somerville, NJ, USA)	Hemostatic forceps and hemostatic clips (HX-610-135 or HX-610-090L; Olympus, Tokyo, Japan)	7 days
Jung, 2021 [53]	Korea	RCT	multiple	143	High risk	<2	Polysaccharide hemostatic powder (EndoClot; Endo-Clot Plus, Inc., Santa Clara, CA, USA)	Hemostatic forceps and hemostatic clip	4 weeks
Kantsevoy, 2014 [37]	USA	Retrospective case series	single	4	NA	10.00 ± 5.80	Endoscopic overstitch suturing (Apollo Endosurgery Inc., Austin, TX, USA)	NA	3 months
Kataoka, 2019 [10]	Japan	RCT	multiple	137	High risk	25.50 ± 15.00	PGA (Neoveil; Gunze Co., Osaka, Japan) + fibrin glue (Beriplast P Combi-Set; CSL Behring Pharma, Tokyo, Japan); step-by-step method, clip-and-pull method	Coagulation using hemostatic forceps	28 days
Kawata, 2018 [45]	Japan	Retrospective cohort study	single	105	High risk	21.00 ± 10.41	PGA (Neoveil; Gunze, Kyoto, Japan) + fibrin glue (Beriplast P Combi-Set; CSL Behring Pharma, Tokyo, Japan); original method	Coagulation using hemostatic forceps	≥20 days
Kikuchi, 2019 [46]	Japan	Retrospective cohort study	single	123	High risk	NA	PGA (Neoveil; Gunze Co., Kyoto, Japan) + autologous fibrin glue; clip-and-pull method	Coagulation using hemostatic forceps	8 weeks
Kobayashi, 2021 [47]	Japan	Retrospective case series	single	24	High risk	10.50 ± 6.70	Wafer paper and ring-mounted PGA sheet (WaRP)	NA	≥17 days
Lee, 2011 [38]	Korea	RCT	single	52	Low risk	17.08 ± 6.24	Detachable snare and clips (Olympus, Tokyo, Japan)	Mucosal defects unclosed	8 weeks
Maekawa, 2015 [39]	Japan	Prospective, single-arm study	single	12	NA	15.18 ± 7.64	Combined use of a single over-the-scope clip (OTSC [Ovesco Endoscopy, Tübingen, Germany]) and through-the-scope clips (TTSCs, ZEOCLIP [Zeon Medical Inc., Tokyo, Japan] or Rotatable Clip Fixing Device, EZ Clip, long type, HX-610135L [Olympus Medical Systems Corp., Tokyo, Japan])	NA	2 months
Mori, 2018 [48]	Japan	RCT	single	39	Low risk	27.20 ± 18.10/35.98 ± 12.38	PGA (Neoveil; Gunze Co., Kyoto, Japan) + fibrin glue (Beriplast P combi-set; CSL Behring Pharma, Tokyo, Japan) + device delivery station system (DDSS)	PGA (Neoveil; Gunze Co., Kyoto, Japan) + fibrin glue (Beriplast P combi-set; CSL Behring Pharma, Tokyo, Japan)	7 days
Nishiyama, 2022 [40]	Japan	Prospective, single-arm study	single	48	High risk	29.90 ± 12.50	O-ring nylon loop and hemoclip (E-LOC) (HX-610- 090; Olympus, Tokyo, Japan)	NA	12–13 days
Pioche, 2016 [18]	France	Retrospective case series	multiple	19	Mixed	2.10 ± 1.20	Self-assembling peptide gel (PuraStat; 3-D Matrix Ltd., Tokyo, Japan)	NA	1 months
Subramaniam, 2019 [54]	UK	Prospective, single-arm study	single	11	Mixed	NA	self-assembling peptide gel (PuraStat; 3-D Matrix Ltd., France)	Coagulation using knife or snare tip using forced/swift coagulation or coagrasper in soft coagulation mode	1 months
Shiotsuki, 2021 [41]	Japan	Retrospective cohort study	single	178	High risk	20.75 ± 9.17	Endoloop (HX-20Q-1, MAJ-340, MAJ-254; Olympus, Tokyo, Japan) and Endoclips (HX-110LR, HX-610; Olympus, Tokyo, Japan)	Coagulation using hot biopsy forceps	2 months
Tan, 2016 [19]	Malaysia	Retrospective cohort study	single	397	Low risk	15.25 ± 28.95	Fibrin glue (YueLingJiao, Hangzhou PuJi Medical Tech, Hangzhou, China)	Coagrasper or hemostatic clips (Olympus, Tokyo, Japan)	12 months
Tsuji, 2015 [49]	Japan	Nonrandomized trial with historical control	single	86	High risk	20.40 ± 9.50	PGA (Neoveil; Gunze Co., Kyoto, Japan) + fibrin glue (Beriplast P Combi-Set; CSL Behring Pharma, Tokyo, Japan); clip-and-pull method	Coagulation using hemostatic forceps in soft coagulation mode	≥14 days
Uraoka, 2016 [58]	Japan	Prospective, single-arm study	single	51	Mixed	<1	Self-assembling peptide gel (PuraStat; 3-D Matrix Ltd., Tokyo, Japan)	NA	8 weeks
Wang, 2020 [55]	China	Retrospective cohort study	single	230	Low risk	NA	Fibrin sealant (BIOSEAl; Guangzhou Bioseal Biotechnology Co., Ltd., Guangzhou, China)	Coagulation using hot biopsy forceps	1 months
Yoshida, 2021 [42]	Japan	Retrospective, single-arm study	single	10	Low risk	6.5 ± 15.27	Part of the S-O clip (Zeon Medical, Toyama, Japan) +open–close SureClip clips (Microtech, MI, USA) +endoclips (HX-610-090S, HX-610-090, HX-610-090L, HX-610-135L; Olympus, Tokyo, Japan) (LOCCM)	NA	2 months
Yu, 2022 [56]	China	Retrospective cohort study	multiple	270	Mixed and grouped	1.80 ± 0.43	Polyethylene oxide adhesive (EndoClot; EndoClot Plus Co., Ltd., Suzhou, Jiangsu, China)	Coagulation using hemostatic forceps	15 days
Zhang, 2013 [57]	China	RCT	single	110	Low risk	4.97 ± 24.27	α-cyanoacrylate medical adhesive (COMPONT; Beijing Compont Medical Devices Co., Ltd., Beijing, China)	Coagulation using APC or hot biopsy forceps	12 months

SD, standard deviation; PGA, polyglycoloc acid; NA, not available; RCT, randomized controlled trial.

## Data Availability

Not applicable. No new data were created in this meta-analysis.

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
