# Peer review of "Endoscopic Delivery of Polymers Reduces Delayed Bleeding after Gastric Endoscopic Submucosal Dissection: A Systematic Review and Meta-Analysis"

_polymers, 2022, doi:10.3390/polym14122387_

Round 1

Reviewer 1 Report

Thank you for the opportunity to review this manuscript. The authors of this manuscript have studied this interesting paradigm which has previously been studied only in small sample sizes. They have done a good job in compiling the data. However, do the cumulative effects of confounders from multiple small studies pose a risk for introducing bias in the analysis. Especially in case of multiple comparisons as in this analysis. Overall, this is a well written manuscript but I would strongly encourage the authors to add correction to the analysis for multiple comparisons to reduce the risk of type I error.

Author Response

Thank you for your kind remarks. We appreciate your raising the question of correcting for multiple comparisons in (network) meta-analysis to manage inflated Type I error.

After extensive literature search, however, we have not found correction for significance levels done in any of the clinical (network) meta-analyses. One possible explanation is that frequently used significance correction methods (e.g., Bonferroni, Holm, Hochberg, Hommel, Benjamini & Hochberg [“False Discovery Rate”], Benjamini & Yekutieli, and etc.) were designed for frequentist p-values calculated by the frequentist method. However, in the Bayesian framework used in our network meta-analysis, likelihood was calculated with arm-based data using Markov chain Monte Carlo simulation (5,000 burn-in and 20,000 iterations), which produce 95% credible intervals for significance interpretation (Shim et al., 2019). Additionally, an existing study showed that Type I error is controlled in the Bayesian network meta-analysis (Uhlmann et al., 2018). Therefore, we retained our original 95% credible intervals for significance interpretation.

SHIM, S. R., KIM, S.-J., LEE, J. & RüCKER, G. 2019. Network meta-analysis: application and practice using R software. Epidemiology and health, 41, e2019013-e2019013.

UHLMANN, L., JENSEN, K. & KIESER, M. 2018. Hypothesis testing in Bayesian network meta-analysis. BMC Medical Research Methodology, 18, 128.

Reviewer 2 Report

Based on a comprehensive literature search, the authors performed a detailed meta-analysis of 31 studies evaluating delayed bleeding rates in patients undergoing gastric endoscopic submucosal dissection (ESD) treated with subsequent endoscopic management, including endoscopic closure, polyglycolic acid (PGA) sheet shielding and hemostatic powder/gel spray.  After systematic review of available evidence, they conclude that endoscopic PGA shielding methods are beneficial for the prevention of delayed bleeding after gastric ESD, especially in high-risk patients, while hemostatic spray would suffice for anti-bleeding management in low-risk patients.

This review article presents a very well conducted and written systematic review and meta-analysis of the clinically important prevention of post-ESD delayed bleeding in gastric lesions that helps shed light on sometime inconclusive and controversial small studies.

Line 29, and throughout the article: Change “systemic” with “systematic”.

Table 1. Add a column for the total number of patients in each study.  Define acronyms at the bottom of the table.

Fig. 2. Regarding the risk ratio plots, indicate in the legend what do the size of the gray squares and the black line (95% confidence interval) represent.

Fig. 3. Regarding the network graph, describe in the legend how to interpret the information in this figure. Enlarge the blue text labels (above the blue dots) that are not legible. Indicate what the darkness of the blue lines represent.

Fig. 4. The second and third columns of risk ratio plots need to be labeled B and C.

Suppl. Fig. 8. Indicate the unit of the Y axis (0.0 -1.0).  Indicate what the 4 color coded columns represent.

Suppl. Fig. 9. Label the x and y axis with the name of the variables.

Author Response

Thank you for commenting on our work. We greatly appreciate your suggestions.

1. Wording has been adjusted throughout the manuscript.

2. We have added a column of sample size in Table 1 and defined acronyms at the bottom of the table.

3. Figure legend has been added in Figure 2.

4. Label sizes have been enlarged in Figure 3, and figure legend has been expanded.

5. Figure 4 is now properly labeled.

6. We have now supplemented Y axis range in the figure legends of Supplementary Figure 8, and have added legend in the figure regarding meanings of color-coded columns.

7. Axis labels have been added to Supplementary Figure 9.

Reviewer 3 Report

The presented review reflects all the main trends on the topic, shows the method of analysis of literary sources. The review is well structured and justified